# Inhibition of Carbonic Anhydrase IX Promotes Apoptosis through Intracellular pH Level Alterations in Cervical Cancer Cells

**DOI:** 10.3390/ijms22116098

**Published:** 2021-06-05

**Authors:** Ebru Temiz, Ismail Koyuncu, Mustafa Durgun, Murat Caglayan, Ataman Gonel, Eray Metin Güler, Abdurrahim Kocyigit, Claudiu T. Supuran

**Affiliations:** 1Program of Medical Promotion and Marketing, Health Services Vocational School, Harran University, Sanliurfa 63300, Turkey; 2Department of Medical Biochemistry, Faculty of Medicine, Harran University, Sanliurfa 63290, Turkey; ismailkoyuncu1@gmail.com (I.K.); atamangonel@gmail.com (A.G.); 3Department of Chemistry, Faculty of Arts and Sciences, Harran University, Sanliurfa 63290, Turkey; 4Department of Medical Biochemistry, Diskapi Yildirim Beyazit Training and Research Hospital, Ankara 06110, Turkey; drmuratcaglayan@gmail.com; 5Department of Medical Biochemistry, Faculty of Hamidiye Medicine, University of Health Sciences Turkey, Istanbul 34668, Turkey; eraymetinguler@gmail.com; 6Department of Medical Biochemistry, Faculty of Medicine, Bezmialem Vakif University, Istanbul 34093, Turkey; akocyigit@bezmialem.edu.tr; 7NEUROFARBA Department, Section of Pharmaceutical and Nutriceutical Sciences, Università degli Studi di Firenze, Sesto Fiorentino, 50019 Florence, Italy

**Keywords:** CAIX, sulphonamide, cell death, cervical cancer, anti-cancer agent

## Abstract

Carbonic anhydrase IX (CAIX) is a hypoxia-related protein that plays a role in proliferation in solid tumours. However, how CAIX increases proliferation and metastasis in solid tumours is unclear. The objective of this study was to investigate how a synthetic CAIX inhibitor triggers apoptosis in the HeLa cell line. The intracellular effects of CAIX inhibition were determined with AO/EB, AnnexinV-PI, and γ-H2AX staining; measurements of intracellular pH (pHi), reactive oxygen species (ROS), and mitochondrial membrane potential (MMP); and analyses of cell cycle, apoptotic, and autophagic modulator gene expression (Bax, Bcl-2, caspase-3, caspase-8, caspase-9, caspase-12, Beclin, and LC3), caspase protein level (pro-caspase 3 and cleaved caspase-3, -8, -9), cleaved PARP activation, and CAIX protein level. Sulphonamide CAIX inhibitor E showed the lowest IC_50_ and the highest selectivity index in CAIX-positive HeLa cells. CAIX inhibition changed the morphology of HeLa cells and increased the ratio of apoptotic cells, dramatically disturbing the homeostasis of intracellular pHi, MMP and ROS levels. All these phenomena consequent to CA IX inhibition triggered apoptosis and autophagy in HeLa cells. Taken together, these results further endorse the previous findings that CAIX inhibitors represent an important therapeutic strategy, which is worth pursuing in different cancer types, considering that presently only one sulphonamide inhibitor, SLC-0111, has arrived in Phase Ib/II clinical trials as an antitumour/antimetastatic drug.

## 1. Introduction

Cancer is an important public health problem worldwide with few effective treatment options, poor prognosis, and high mortality rates [1]. Despite great discoveries in cancer treatment, cancer-related mortality rate increases day by day [2]. In addition, traditional methods (chemotherapy, radiotherapy, and surgical interventions) used in cancer treatment have side effects that negatively affect the quality of life. Therefore, more rigorous studies are required to develop more specific and effective treatment strategies [3,4].

The main reason underlying the resistance mechanism against chemotherapy and radiotherapy is hypoxia and acidosis caused by disturbed pH in the cancer cell [5]. Solid tumours induce hypoxia due to malnutrition and low oxygen supply [6]. To survive in these conditions, cancer cells must adapt. The most important transcription factor in hypoxia is the hypoxia-inducible factor-1 (HIF-1) [7]. HIF-1 has been heavily implicated in the regulations of epithelial–mesenchymal transition (EMT), glycolysis, migration, pH regulation, and angiogenesis [8,9,10]. As cancer cells have a high rate of metabolic activity, they use both the oxidative phosphorylation of glucose and its anaerobic glycolysis even in normoxic condition for energy production, which leads to increased production of metabolic acids [11]. These metabolic byproducts are removed from cells to prevent the accumulation of intracellular H^+^ ions, and the cells are kept at an alkaline pH, through the intervention of a multitude of proteins [12]. 

Tumour cells can maintain pHi values by increasing expression and activation of pH-regulating proteins, some of which are HIF-1-dependent, such as the monocarboxylate transporter-4 (MCT-4), carbonic anhydrase IX (CAIX), sodium bicarbonate transporters, and sodium-proton exchangers, which release lactate, CO_2_/bicarbonate, and H^+^ from tumour cells. This abnormal regulation of H^+^ ions concentrations leads to an inverted pH gradient, which has become a phenomenon increasingly regarded as one of the distinctive hallmarks of cancer cells [13].

CAIX is an enzyme that is overexpressed in hypoxia, being involved in tumorigenesis, proliferation, and invasion of cancer cells [14]. It catalyses the reversible hydration of CO_2_ to bicarbonate ions and protons [15]. Overexpression of CAIX in tumour cells promotes survival in hypoxia [16] and prompts cell migration in cervical cancer lines, as demonstrated for SiHa cells, among others [17]. On the other hand, many studies indicated that inhibiting this enzyme has a therapeutic effect on primary tumour and metastases growth [18,19]. These findings were relevant for the validation of CAIX as an anticancer therapeutic target [20,21]. 

Sulphonamide derivatives acting as effective CAIX inhibitors showed significant anticancer activity in vitro and/or in vivo in many cell types and cancer models, with one such compound, SLC-0111, presently in Phase Ib/II clinical trials for the management of advanced, metastatic solid tumours [18,21]. Small-molecule CAIX inhibitors [21], but also small-molecule drug conjugates, antibody-drug conjugates, or cytokine-drug conjugates, targeting these enzymes were demonstrated to constitute an innovative antitumor strategy in the last decade [22]. In the present study, the cytotoxic effects of a sulphonamide CAIX inhibitor, derivative compound **E** (4-(2-((5-bromo-2-hydroxybenzyl) amino) ethyl) benzenesulphonamide), were investigated in the cancer cells with high CAIX expression.

## 2. Results

### 2.1. Compound E Decreased Viability of Cell Lines

Dose- and time-dependent cytotoxic effects of compound **E** on cancer (HT-29, HeLa, MDA-MB-231) and normal cells (HEK-293 and PNT-1A) were investigated using the WST-1 assay. When the effect of compound **E** on the cell viability was evaluated, the lowest IC_50_ and the highest selective index (SI) values were found in HeLa cells (Table 1). 

### 2.2. Compound E Reduced Proliferation of HeLa Cells 

The effects of compound **E** on HeLa cell proliferation were investigated using the BrdU ELISA method. Compound **E** was found to reduce the proliferation of HeLa cells in a dose-dependent manner (Figure 1).

### 2.3. Compound E Triggered Apoptotic Morphology in HeLa Cells 

Acridine orange-ethidium bromide (AO/EB) is a staining method indicating the apoptotic cells. Acridine orange stains both living and dead cells, whereas ethidium bromide stains only cells with impaired membrane integrity. While the live cells are stained green, the early apoptotic cells are stained bright green-yellow, showing nuclear fragmentation. Furthermore, cells in late apoptotic and necrotic periods are stained with ethidium bromide in orange-red colour. In addition to staining, apoptotic markers can be morphologically detected. In our study, we observed both markers of apoptosis in HeLa cells treated with different doses of compound **E** (10, 25 and 50 µM), and the observed effects become more apparent with increasing doses (Figure 2).

### 2.4. Compound E Increase Apoptotic Cell Ratio in HeLa Cells 

Annexin V/PI staining was performed to determine the ratio of dead cells. It was recorded that a dose of 25 µM compound **E** indicated a dramatic apoptotic effect, and the early apoptotic cell ratio was increased (Figure 3). Moreover, the degree of this apoptotic effect was positively correlated with the dose of compound **E**.

### 2.5. Compound E Disrupted pH Regulation of HeLa Cells

CAIX is a transmembrane protein that regulates intracellular and extracellular pH in response to hypoxia. The effect of compound **E** on intracellular pHi was measured using a fluorescent BCFL-AM indicator. We found that compound **E** decreased intracellular pHi level after the dose of 25 µM in a dose-dependent manner (Figure 4).

### 2.6. Compound E Increased Intracellular ROS Level in HeLa Cells 

The excess intracellular H^+^ and intracellular free radical (ROS) are markers of malignant proliferation. The relationship between intracellular pHi and intracellular free radical (ROS) is also critical in the regulation of apoptotic mechanism. When the effects of compound **E** in HeLa cells were examined, it was found that the amount of ROS increased in a manner dose-dependent on compound **E** (Figure 5). Increased intracellular H^+^ concentration and ROS amount triggered apoptosis.

### 2.7. Compound E Disrupted Mitochondrial Membrane Potential in HeLa Cells 

Mitochondrial membrane potential is critical for maintaining the physiological function of the respiratory chain in cells. It is known that disruption of the membrane integrity causes activation of the apoptotic pathway by increasing the production of free radicals as well as disrupting the energy production mechanism. In this study, we found that the sulphonamide **E** disrupts mitochondrial membrane potential in HeLa cells and thus increases intracellular free radical level (Figure 6). 

### 2.8. Sulphonamide E Arrested Cell Cycle in HeLa Cells

PI staining was used to determine the cell cycle phases. HeLa cells were analysed after incubating with the compound **E** at doses of 10, 25, and 50 µM for 72 h and confirming the presence of 10.000 cells in flow cytometry. The obtained results are as follows. WT: G0/G1 = 56.6%, S = 23.2%, G2/M = 20.4%; 10 µM: G0/G1 = 56.8%, S = 19.3%, G2/M = 22.3%; 25 µM: G0/G1 = 66.5%, S = 14.6%, G2/M = 17.9% and 50 µM: G0/G1 = 68.8%, S = 10.8%, G2/M = 19.3%. Compound **E** arrested cell cycle of HeLa cells at the G0/G1 phase (Figure 7).

### 2.9. Compound E Triggered DNA Damage in HeLa Cells

Immunofluorescence staining of γ-H2AX level was performed to determine the effects of the compound **E** on DNA damage in HeLa cells. This method is used to provide primary detection of DNA damage caused by free radical and UV radiation. According to the staining results, the compound **E** induced DNA damage in a dose-dependent manner (Figure 8). 

### 2.10. Sulphonamide E Increased the Expression of Apoptosis-Related Genes in HeLa Cells

Expression levels of apoptotic genes were evaluated using mRNA samples of HeLa cells treated with compound **E** at a dose of 25 µM. The Ct values were normalized with the GAPDH gene. The expression changes were analysed by calculating 2^-ΔΔCt (fold change). The expression levels of CASP-3,8,9 and 12 genes increased at least two-fold in the group treated with 25 µM compound **E** compared to the control group (WT). While the level of pro-apoptotic BAX gene expression significantly increased, the change in BCL-2 gene was not significant. The expression levels of NRF-2, and BECLIN-1 and LC3 genes, mediators of autophagy pathway, increased, suggesting that type I and type II death pathways were triggered in HeLa cells after the compound **E** administration (* *p* < 0.05; ** *p* < 0.01) (Figure 9).

### 2.11. Compound E Increased the Level of Apoptosis-Related Proteins in HeLa Cells

After compound **E** administration, protein levels of the apoptotic mediator genes CASP-3, 8, 9 and cleaved PARP genes were studied by Western blot method in HeLa cells. Furthermore, CAIX protein level was examined to test the inhibitor efficacy of the compound **E**. When the group treated with different doses of the compound **E** and WT were compared, pro-caspase 3 protein level was found to be decreased, while cleaved caspase 3, 8, and 9 protein levels were found to be increased. Furthermore, the compound **E** treatments of 10 and 25 µM doses increased PARP protein level in the treatment group compared to WT. Moreover, the compound **E** of 25 µM dose was found to be effective for inhibiton of CAIX protein (Figure 10).

## 3. Discussion

CAIX inhibition has been validated recently as a new approach for targeting hypoxic tumours, with at least one compound in Phase Ib/II clinical trials [18,20,21,22,23]. Understanding the role of CAIX at various stages of cancer development (e.g., metabolic transformation, growth, invasion, and metastasis) and investigating functional and biomolecular changes associated with CAIX inhibition are critical for drug development [24,25]. As mentioned above, hypoxia-induced CAIX plays a role in intracellular and extracellular pH regulation [26], whereas CAIX is ectopically expressed by most hypoxic tumour types in advanced stages of the disease [7].

In this study, we investigated the therapuetic effects of the sulphonamide CAIX inhibitor **E**, a sulfonamide derivative developed to inhibit CAIX, in HeLa cells, which show high CAIX expression. With functional and molecular analyses, we showed that pHi level significantly decreased with CAIX inhibition in HeLa cells under normoxic conditions, which triggered the intrinsic and extrinsic apoptotic pathways. Recent studies demonstrated that CAIX expression increases in breast [16], colon [27], cervix [15], melanoma [28], and lung [25] cancers. Thus, CAIX inhibition with small molecule derivatives as a treatment strategy of cancer has been intensely investigated [20,21,23,29]. 

In this regard, here, we incubated MDA-MB-231, HeLa, and HT-29 cells, which are known to have high CAIX expression, with different concentrations of sulphonamide **E** for 24, 48, and 72 h. According to cell viability assay, we detected the lowest IC_50_ (20.1μM) and the highest SI (>3.7) value in the HeLa cells (5-Fu (14.7 µM)) (Table 1). Moreover, according to the BrdU study result, we found that the proliferation of HeLa cells significantly decreased (Figure 1), which is consistent with the findings of Tülüce et al. [27,30], showing that sulphonamide derivatives structurally related to compound **E** and acting as CAIX inhibitors triggered apoptosis by reducing cell proliferation. Güttler et al. [16] showed that reducing CAIX expression in breast cell lines (MDA-MB-231 and MCF-7) using siRNA transfection and CA inhibitor (CAI) U104 triggered apoptosis by increasing intracellular H^+^ concentration. Upon inhibition of CAIX with compound **E**, we observed the same morphological signs of apoptotsis in addition to gene and protein analyses. Our AO/EB staining indicated substantial morphological changes with the inhibition of CAIX by compound **E** (Figure 2), which is consistent with the findings of cell death, cell cycle, gene and protein analyses. However, the inhibited CAIX-driving effects was not tested by using rodent models, which is the main limitation for corroborating the cell line findings.

Koyuncu et al. [14,15] showed that CAIX inhibition causes morphological changes in cells and increases apoptotic cell rate in a manner dependent on inhibition of CAIX. When testing these reported morphological findings induced by CAIX inhibition with AnnexinV-FITC/PI staining, we found that the ratio of apoptotic cells dramatically increased upon CAIX inhibition by the compound **E** in HeLa cells compared to the control cells (Figure 3). 

Cellular acidosis triggers the early stage of apoptosis, and therefore the intracellular H^+^ hemostasis is kept in a narrow range to prevent apoptosis [31,32]. The decrease in pHi causes DNA damage, MMP disruption and thus ROS accumulation, resulting in apoptosis. Preventing this devastating cascade have been heavily implicated in recent studies, and studies were designed to investigate combined treatment strategies targeting the pHi/ROS/MMP for solid tumour therapy [33,34,35].

When evaluating the effects of compound **E** on intracellular pH level, we found that pHi decreased (Figure 4), ROS increased (Figure 5), and MMP (Figure 6) was disrupted in a manner dependent on the dose of the compound **E**. These results suggest that the apoptotic mechanism was triggered by compound **E** administration through pH regulation, MMP disruption, and ROS accumulation. Intracellular ROS accumulation was indicated to induce DNA damage and subsequently arrest the cell cycle at G1 phase [36,37,38]. To corroborate these apoptotic effects, we stained the cells with PI and γ-H2AX and found the compound **E** (25 μM) arrested the cell cycle at the G0/G1 (66.5%) phase and increased DNA damage (Figure 7 and Figure 8).

Caspase activation initiates cellular degradation [39]. Disrupted MMP increases intracellular ROS level [40], causes DNA damage [41], and subsequently activates caspases. The activated intrinsic apoptosis is critical in the initiation of ROS-mediated cell death [41,42,43]. Disruption of the mitochondrial membrane integrity leads to the activation of a number of proteins in the intermembrane space of mitochondria, resulting in alterations of BAX and BCL-2 genes expressions and release of cytochrome c [44]. Cytochrome c activates caspase-3 by forming an apoptosome complex consisting of cytochrome c/apoptotic protease activating factor 1 (APAF-1)/caspase-9 [45].

To evaluate the expression of genes involved in the apoptotic pathway triggered by CAIX inhibiton, we investigated the expression of genes regulating the apoptotic pathway (CAS-3,8,9,12, BAX and BCL-2) and found that caspases and BAX significantly increased but BCL-2 did not change (* *p* < 0.05; ** *p* < 0.01) (Figure 9), which is consistent with the findings of Western blot. However, the protein level of pro-caspase-3 decreased, and the significant increases in protein levels of cleaved 3, 8, and 9 were recorded. Furthermore, the protein level of cleaved PARP, an indicator of DNA damage, increased (Figure 10). 

In addition to apoptosis, we found that the expression levels of the autophagy, BECLIN-1 and LC3 genes, and ROS markers, NRF-2 gene, increased, suggesting that apoptosis and type-two cell death pathway were activated in HeLa cells (Figure 9).

We investigated the effects of compound **E**, which was designed as a potent CAIX inhibitor, on cell survival and cellular function. Based on the findings, we suggest that in HeLa cells with high CAIX expression, inhibiting CAIX with compound **E** triggers pHi/ROS/MMP pathway by increasing intracellular H^+^ concentration and arrests cell proliferation. This study supports the general consensus about inhibition of CAIX in the relevant literature and proposes specific CAIX inhibitors as therapeutic agents for solid and aggressive cancers.

## 4. Materials and Methods

### 4.1. Materials

We purchased the cell-culture medium (RPMI 1640), fetal bovine serum (FBS), streptomycin, and penicillin, from Gibco BRL (Life Technologies, Paisley, Scotland); DMEM-F12, WST-1 (Roche, Basel, Switzerland), ethidium bromide, acridine orange, 2,7- dichlorofluorescein diacetate (DCFH-DA), trypsin-EDTA solution, and dimethyl sulfoxide (DMSO), from Sigma-Aldrich Chemical Company (Germany); and the culture plates from Nunc (Brand products, Denmark).

### 4.2. Cell Culture 

We used cancer and normal cell lines purchased from ATCC (American Type Culture Collection) and ECACC (European Collection of Authenticated Cell Cultures) and stored in liquid nitrogen. HT-29 (ATCC: HTB-38, colon adenoma cancer), HeLa (ATCC: CCL-2, cervix adenoma cancer cell), MDA-MB-231 (ATCC: HTB-26, breast adenoma cancer cell), and HEK-293 (ATCC: CRL-1573, embryonic kidney epithelial cell) cell lines were incubated in DMEM-F12 (Sigma-Aldrich, Germany), including 10% Fetal Bovine Serum (FBS), 100 μg/mL streptomycin/100 IU/mL penicillin, at 37 °C in an incubator containing 5% CO_2_, 95% air in a humid atmosphere. PNT-1A (ECACC: 95012614, normal prostate cells) cell line was incubated in RPMI-1640 medium, including %10 FBS, 100 μg/mL streptomycin/100 IU/mL penicillin, at 37 °C in an incubator containing 5% CO_2_, 95% air in a humid atmosphere.

### 4.3. Sulphonamide Derivative E

The synthesis, characterization, spectral, and analytical data of the sulphonamide derivative used in this study were reported in the previous study by our group [46,47]. Sulphonamide **E** was re-synthesized and characterized as previously described [46,47]. Inhibition data for this compound; 4-(2-((5-bromo-2-hydroxybenzyl) amino) ethyl) benzenesulphonamide (**E**) (colour: White; M.P. 189–191 °C; C_15_H_17_BrN_2_O_3_S (385.28 g/mol)) against relevant CA isoforms are shown in Figure 11. Sulphonamide **E** shows effective inhibitory properties against several CA isoforms (Figure 11), as measured using a stopped-flow assay and the physiological reaction catalysed by these enzymes, CO_2_ hydration to bicarbonate, and protons [48].

### 4.4. Cytotoxicity Analysis

The cytotoxic effects of compound **E** were evaluated with WST kit (Roche, Basel, Switzerland) in accordance with manufacturer protocols. The cells were planted in 96 well plates (10^4^ cells per well). After 24 h of incubation, compound **E** at doses of 0, 2.5, 5, 10, 25, 50, 100, and 200 μM was administrated on the cell lines for 24, 48, and 72 h, and 5-FU on the control cell line with the same doses. WST-1 reactive of 10 µL was added in all wells. Following 4 h incubation, the measurements were carried out in a plate reader (Spectramax M5, BioTek, VT, USA) at wavelengths of 450 and 630 nm. Consequently, graphs were created, and the IC_50_ values of compound **E** and 5-fluorouracil were calculated. Selectivity index (SI) can be defined as the ratio of the toxic concentration of a sample against its effective compound concentration.

### 4.5. Investigation of Antiproliferative Effects 

The effects of compound **E** on the proliferation of HeLa cells were investigated by a commercial proliferation kit BrDU (BioVision, CA, USA) according to the protocols of the manufacturer. Twenty-four hours later, the cells were planted in 96 well plates (10^4^ cell/mL), the medium was replaced, and the substance was administrated at a range from 2.5 to 200 μM for 72 h. Subsequently, the media were added, after which 100 µL 1X of BrdU reactive was added; then, they were placed in an incubator that contained % 5 CO_2_, 95% fresh air, at 37 °C, for 4 h. The medium was then removed, and 100 μL of fixative/denaturation solution was added and incubated for 30 min. A total of 100 µL 1X of the antibody solution BrdU was added, following 1-h incubation, the solution washed by 300 µL 1X of the washing solution two times. A volume of 100 μL 1X of anti-mouse HRP-linked antibody was added and incubated at room temperature for 1 h. Then, the cells were washed with 300 µL 1X of washing solution. Final incubation was performed by 100 μL of TMB substrate solution for 5 min, after which the stop solution was added; consequently, reading at a wavelength of 450 nm was performed. 

### 4.6. Acridine Orange and Ethidium Bromide (AO/EB) Staining Assay

After 24 h incubation of HeLa cells in 12 well-plates at 5 × 10^4^ cell/mL, the compound **E** at concentrations of 10–50 μM was added on plates, and the cells were incubated at 37 °C for 24 h. The cells were then washed with PBS and incubated in a solution including 100 μL acridine orange (100 µg/mL) and ethidium bromide (100 µg/mL) at room temperature for 5 min. The morphological changes were investigated under fluorescent microscopy (Olympus CKX 51, DP73, USA).

### 4.7. Apoptosis Detection by Annexin V-PI

The apoptotic effects of compound **E** were detected by the commercial FITC Annexin V Apoptosis Detection Kit I (BD Biosciences, NJ, USA) according to the protocol of manufacture. The cells were planted in 6-well plates (5 × 10^5^ cells/well), and following a 24 h incubation period, the compound **E** at concentrations of 10–25–50 μM was administrated and followed by incubation for 24 h. The cells were raised by trypsin and transferred into new tubes within 1X binding buffer and incubated for 15 min at room temperature with 5 μL of fluorochrome-conjugated Annexin V and 5 μL of Propidium Iodide dyes. The cells were involved in 100 uL 1X binding buffer and centrifuged at 1200 rpm for 5 min. Finally, the cells were analysed in flow cytometry (BD Via, NJ, USA).

### 4.8. Detection of Intracellular pH 

Intracellular PH was measured according to the protocol of fluorometric Intracellular pH Assay Kit (Sigma-Aldrich, MAK-150, Germany). Fluorescent BCFL-AM indicator passed through cell membrane to measure intracellular pH fluctuations in the fluorometric intracellular pH kit. BCFL-AM was utilized to measure intracellular pH of the cells. The cells were planted in black plates and incubated for 24 h. The medium was replaced with BCFL-AM Reagent, which had been prepared in 100 µL of HBS solution (Hank’s buffer with 20 mM HEPES, 5 mM Probenecid); then, the cells were incubated at 37 °C with an atmosphere of 5% CO_2_ for 30 min (protected from light). The compound **E** at doses of 0–200 μM was administrated in HBS solution. The measurement was performed at wavelengths 490 nm (excitation) 535 nm (emission) in fluorimetry (Spectramax M5, BioTek, VT, USA.

### 4.9. Detection of Intracellular ROS Production 

Intracellular free radicals were measured using the Cellular Reactive Oxygen Species Detection Assay commercial kit (Abcam-186027, Cambridge, UK). In this kit, a fluorescence probe that is permeable to cells was used. When the probe reacts with ROS, it produces red fluorescence. According to the kit protocol, the cells were planted in black plates and incubated for 24 h. The compound **E** in PBS was administered at doses of 0–200 μM and incubated at 37 °C within an atmosphere of 5 % CO_2_ for 1 h (protected from light). After the compound **E** was poured out, the wells were filled with 100 µL of ROS Deep Red Working Solution, and then they were incubated at 37°C under an atmosphere of 5 % CO_2_ for 1 h. A fluorometry device (Spectramax M5, BioTek, VT, USA) Ex/Em: 650/675 nm (cut off: 665 nm) was used for the measuring.

### 4.10. Detection of Mitochondrial Membrane Potential (MMP) 

Mitochondrial membrane potential (MMP) changes were spectrophotometrically detected. MMP was measured in a fluorometry device (Spectramax M5, BioTek, VT, USA) at wavelengths of 490 nm (excitation) and 530 nm (emission) in suitable conditions according to the manufacturer protocol (Sigma-Aldrich-CS0390, Germany).

### 4.11. Cell Cycle Assay

Cell cycle analysis was performed using BD Cycletest ™ Plus DNA Reagent kit (BD Biosciences, NJ, USA). According to the kit protocol, 1 × 10^6^ cells were planted in six well-plates for 24 h, and the cells incubated for 24 h with the compound **E** at concentrations of 10–25–50 μM. The cells were raised with trypsin and centrifuged at 1500 rpm for 5 min. When a suspension was created by 1X binding buffer, 250 µL solution A was added and incubated in a light-free environment for 10 min, 200 µL solution B was added and incubated in light-free environment for 10 min, and 200 µL solution C was added and incubated at +4 °C in a protected-light environment for 10 min. All these analyses were performed by flow cytometry (BD Via, NJ, USA).

### 4.12. Detection DNA Damage (γ-H2AX) by Immunofluorescence Staining

HeLa cells (1 × 10^4^/^mL^) were seeded in 12-well cell culture plates, and after 24 h incubation, they were treated with the compound **E** at doses of 10–25–50 µM and incubated for 24 h. The cells were washed with PBS and fixed with 100% methanol at −20 °C overnight, which was followed by PBS washing three times for 10 min. The cells were treated with % 0.2 Triton X-100 on a shaker at room temperature for 5 min. It was washed with PBS, and 1% BSA-containing PBS was added and incubated with primary mouse monoclonal anti-γ-H2AX antibody (Cell Signalling Technology, MA, USA) at 37 °C for 1 h. The cells were washed with PBS three times for 5 min. Secondary goat anti-mouse Alexa-488-conjugated IgG (Invitrogen, CA, USA) was administrated and incubated at 37 °C for 20 min, which was followed by PBS washing three times for 4–5 min. One millilitre of 70% EtOH was added, and at +4 °C it was incubated for 5 min. One millilitre of 100% EtOH was added and, on a shaker, incubated at room temperature for 1–2 min. The nucleus was stained with DAPI. The images were recorded by Olympus Inverted fluorescence microscope (Olympus CKX 51, DP73, USA).

### 4.13. RNA Extraction and Real-time Quantitative PCR

All gene expression levels of cells were detected following 24 h incubation with the compound **E** at a dose of 25 μM. Total RNAs were isolated using miRNeasy mini kit (Qiagen Hilden, Germany), and reverse transcription was performed by Ipsogen RT Set (Qiagen Hilden, Germany) according to the kit protocol. RT-qPCR was then performed by QuantiTect SYBR Green PCR kit (Qiagen Hilden, Germany) in Rotor-Gene Q real-time PCR system (Qiagen Hilden, Germany). Each sample was studied in triplicate, using primer sets and GAPDH. Gen expressions were calculated with the 2^-ΔΔCt method to compare to control groups. The GraphPad Prism 8 program was used to calculate *p* values. Primers were designed using Primer blast on the National Center for Biotechnology Information website. All primers were determined to be 95–100% efficient, and all exhibited only one dissociation peak. Sequences are listed in Table 2.

### 4.14. Immunoblot Analysis

The cells were seeded on 6 cm^2^ cell culture dishes and treated with the compound **E** at doses of 10–25–50 μM, for 24 h. The cells were then washed by cold PBS, which was followed by lysis with RIPA lysis tampon (10 mM Tris-HC1 pH:8.1 mM EDTA, 1 mM EGTA, 140 mM NaCl, 1% TritonX-100, 0.1 SDS, 0.1% Sodium deoxycholate), 1X phosphates, and protease inhibitor (Santa Cruz, Heidelberg, Germany). Acquired supernatant was centrifuged at 12,000× g for 10 min and transferred to new tubes. Protein concentration was detected using the protein assay kit BCA (Thermo Fisher Scientific, MA, USA). Following protein (50 µg) standing in 10% SDS-PAGE gel at 50 V for 30 min and 80 V for 3 h, the proteins were blotted to PVDF membrane at 70 V for 5 min (Bio-Rad Turbo Transfer System). After blocking in 1X TBST (12.1 gr Tris, pH 7.5, 70 gr NaCl, %0.1 Tween-20) containing 5% dry milk or 5% BSA, the primer monoclonal antibodies shown in Table 3 were administrated to the membrane overnight, which was followed by washing with 1 X TBS-T and incubation period with secondary antibodies HRP rabbit or mouse (1/10.000) for 60 min. The membrane was washed with 1 X TBS-T again. For band imaging, ECL substrate (EMD Millipore Corp., MA, USA) was used in the imaging system (LI-COR Odyssey Fc, NE, USA).

### 4.15. Statistical Analysis

The distribution of data was calculated using the Shapiro–Wilk test. While Student- t-test was used for the groups with normal distribution (*p* > 0.05), the Mann Whitney U test was used to compare two groups with abnormal distribution (*p* < 0.05). One-way ANOVA test was used to compare three or more groups with a normal distribution. SPSS 25 (SPSS, Inc, CA, USA) and GraphPad Prism 8 (GraphPad Software, Inc, San Diego, CA, USA) were used for the statistical analysis and graph creation (* *p* < 0.05; ** *p* < 0.01).

## Figures and Tables

**Figure 1 ijms-22-06098-f001:**
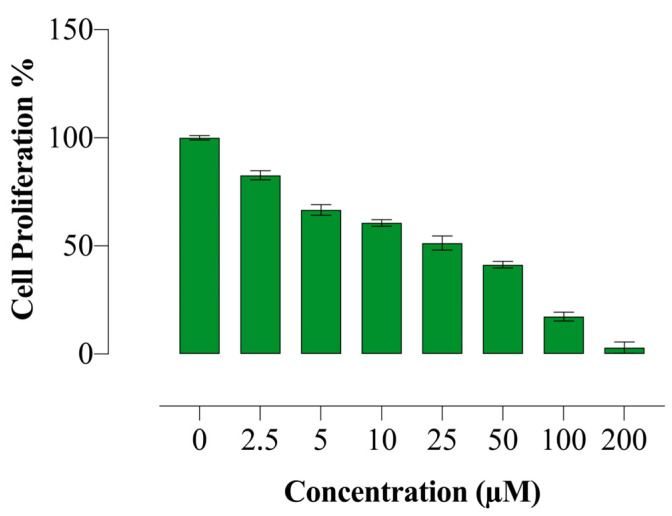
Antiproliferative effect of compound **E** in HeLa cells. Compound **E** decreased proliferation of HeLa cells in a dose-dependent manner. All data are expressed as mean ± SD values from three independent experiments.

**Figure 2 ijms-22-06098-f002:**
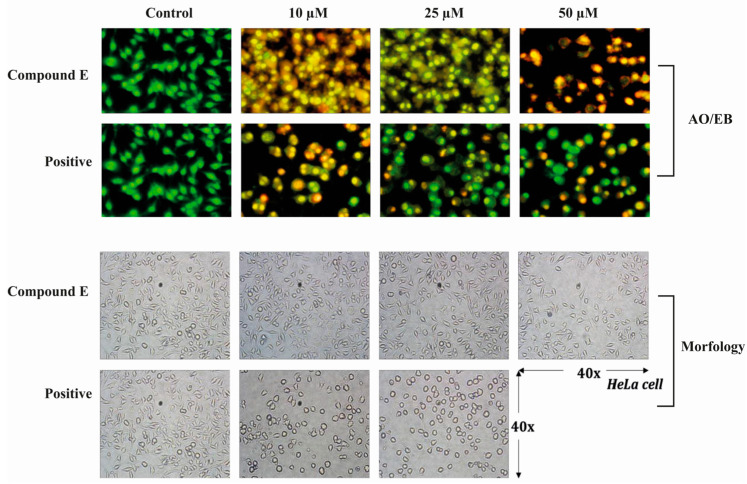
Acridine orange/Ethidium bromide staining of HeLa cells. Compound **E** triggered apoptosis in HeLa cells, which is marked with bright red and yellow, compared to the green-stained control HeLa cells in which compound **E** was not administrated. Images were obtained under florescence microscope.

**Figure 3 ijms-22-06098-f003:**
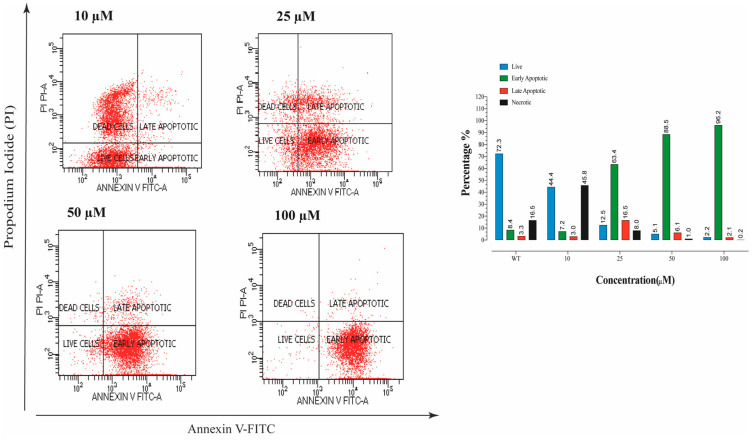
The apoptosis ratio of HeLa cells and contour diagram of Annexin-V/PI staining. A 25 µM effective dose of compound **E** dramatically increased the early apoptotic cell rate up to 63.4% compared to the record in control, 8.4%.

**Figure 4 ijms-22-06098-f004:**
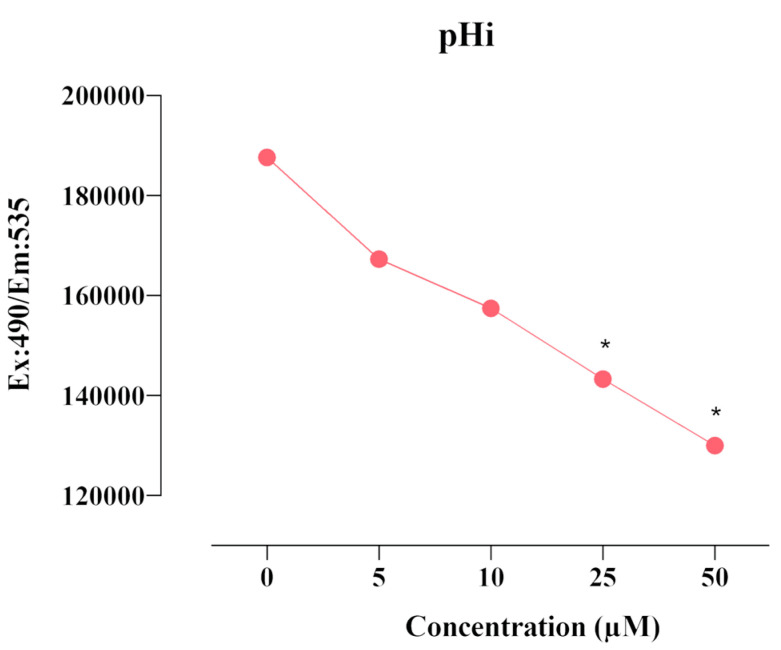
Measurement of pHi level in HeLa cells. A 25 µM dose of compound **E** inhibited CAIX and decreased intracellular pHi level in a dose-dependent manner. The data are presented as the mean ± SD (n = 3). * *p* < 0.05 compared with the control.

**Figure 5 ijms-22-06098-f005:**
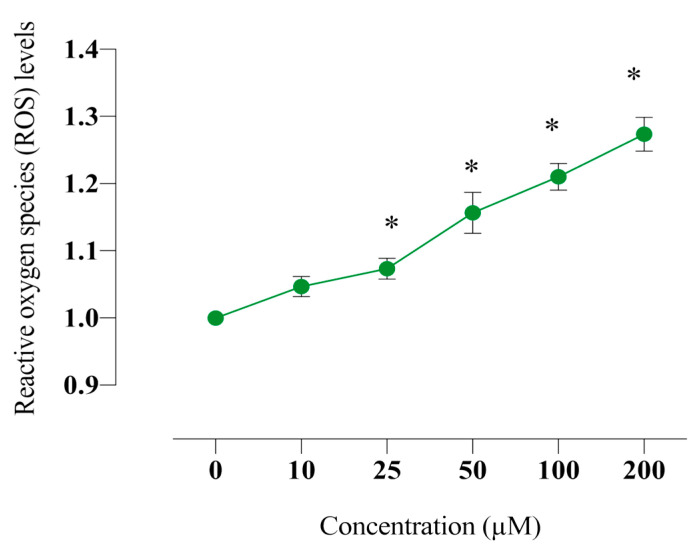
Measurement of intracellular ROS level in HeLa cells. The cells were stained with the ROS red dye, and spectrofluorimetric assay was performed to determine the intracellular ROS. A 25 µM dose of compound **E** inhibited CAIX and increased intracellular ROS level in a dose-dependent manner. The data are presented as the mean ± SD (n = 3). * *p* < 0.05 compared with the control.

**Figure 6 ijms-22-06098-f006:**
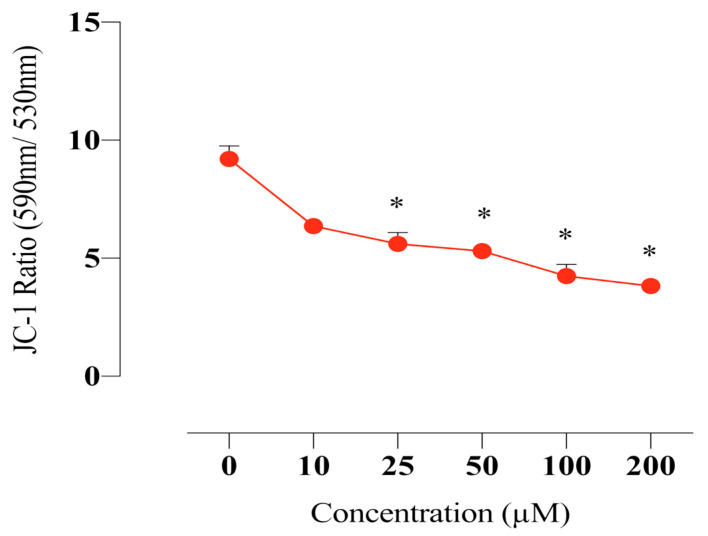
Measurement of the MMP in HeLa cells. The cells were stained with the JC-1 dye, and spectrofluorimetric assay was performed to determine the MMP. A 25 µM dose of compound **E** inhibited CAIX and disrupted MMP in a dose-dependent manner. The data are presented as the mean ± SD (n = 3). * *p* < 0.05 compared with the control.

**Figure 7 ijms-22-06098-f007:**
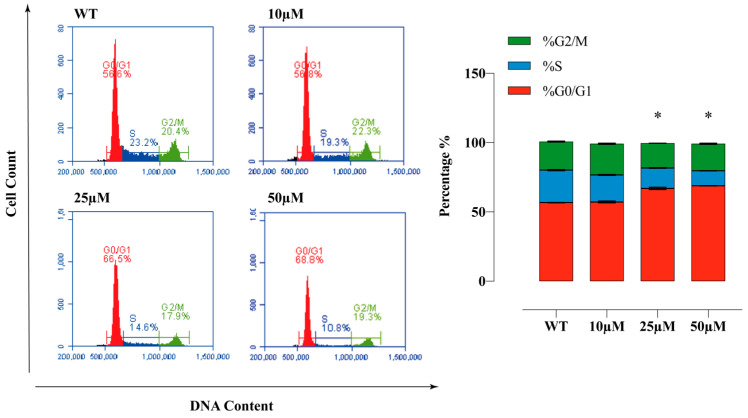
Cell cycle analysis of HeLa cells. The apoptosis ratio of HeLa cells and contour diagram of PI staining. A 25 µM dose of compound **E** inhibited CAIX and dramatically arrested 66.5% of the cells at G0/G1 phase. All data are expressed as mean ± SD values from three independent experiments. * *p* < 0.05 compared with the control (WT: Wild Type).

**Figure 8 ijms-22-06098-f008:**
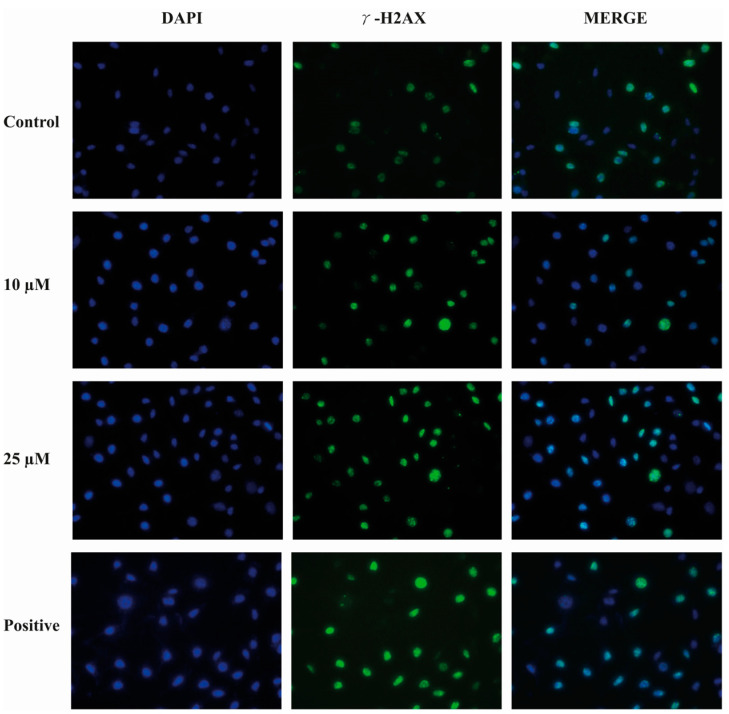
Representative images of DNA damage in HeLa cells. A 25 µM dose of compound **E** inhibited CAIX and apparently increased DNA damage marked with green γ -H2AX staining. Cell fragments were fixed and processed for γ -H2AX immunofluorescent staining (IFA). DAPI blue was used to detect nuclei.

**Figure 9 ijms-22-06098-f009:**
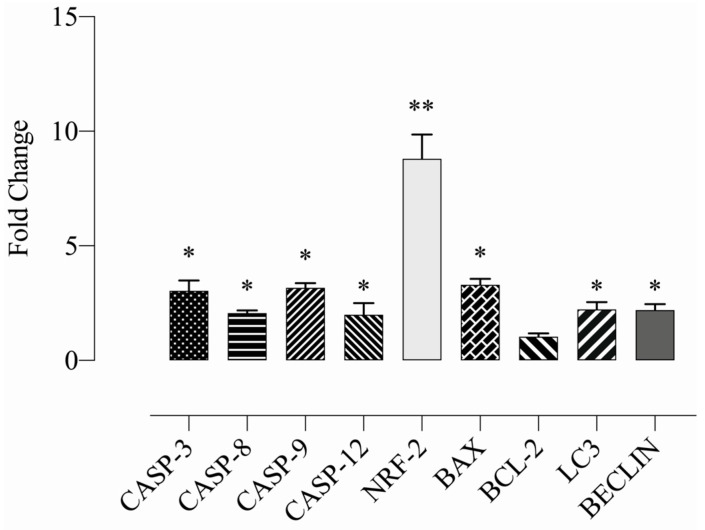
Measurement of the relative apoptotic gene expressions. A 25 µM dose of compound **E**, inhibited CAIX and dramatically increased the expressions of apoptosis (caspase-3, caspase-8, caspase-9, caspase-12 and Bax) and autophagy mediators (LC3, Beclin) and ROS-related genes (NRF-2). The data are presented as the mean ± SD (n = 3). * *p* < 0.05; ** *p* < 0.01 compared with the control.

**Figure 10 ijms-22-06098-f010:**
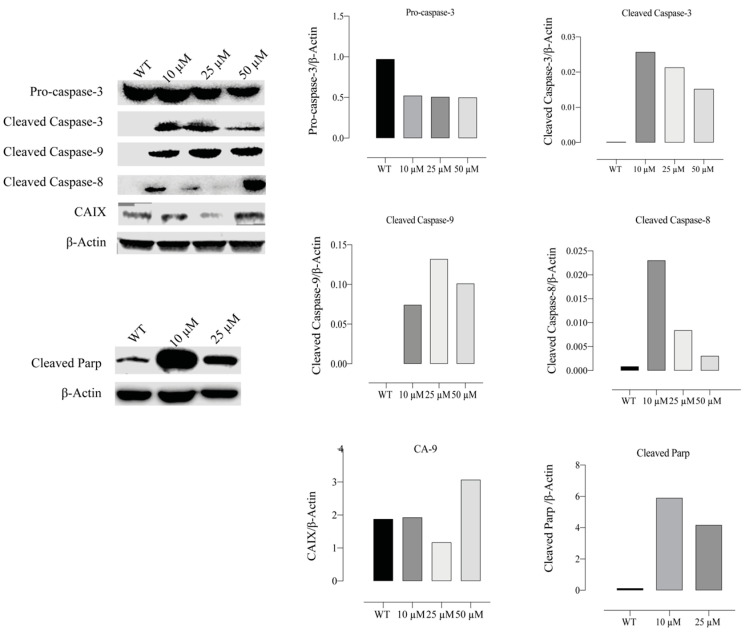
Measurement of apoptotic proteins, caspases 3, 8, and 9. HeLa cells were treated with compound **E** at different doses of 0, 10, 25 and 50 µM. Protein levels of apoptosis were normalized to β-actin. The compound **E** dramatically increased the cleavage levels of caspase 3, 8, and 9 proteins in a dose-dependent manner. The cleavage PARP protein level also increased with the treatment of compound **E** at a dose of 10 µM. The most effective dose of compound **E** on CAIX was found to be 25 µM. The densitometry quantification of the blot was determined by software (Li-Cor Fc). The data are presented as the mean ± SD (n = 3).

**Figure 11 ijms-22-06098-f011:**
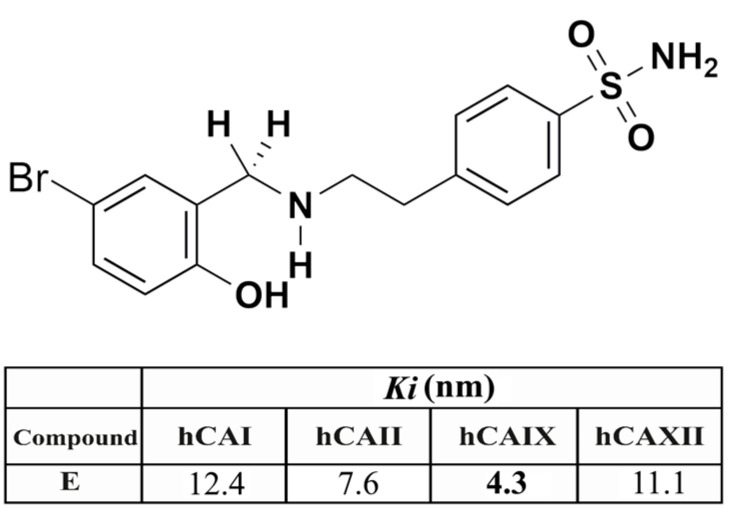
The chemical structure of compound **E** and inhibition data (*K_i_* , nM, values) against the major human (h) CA isoforms I, II, IX, and XII.

**Table 1 ijms-22-06098-t001:** Cytotoxicity of compound **E** and 5-Fluorouracil (5-FU) on cancer and normal cell lines (IC_50_ and SI).

IC50 (µM) ^a^				SI (µM)			
**Compound E**	**24 h**	**48 h**	**72 h**		**24 h**	**48 h**	**72 h**
**HEK-293**	152.0 ± 3.44	296.4 ± 4.67	76.1 ± 2.1	**HEK-293/HELA**	>3.2	>3.4	>3.7
**PNT1-A**	214.1 ± 9.3	167.3 ± 11.4	398.3 ± 10.4	**HEK-293/HT-29**	>1.7	>6.9	0.9
**MDA-MB-231**	1948.4 ± 4.65	205.3 ± 3.55	62.9 ± 10.3	**HEK-293/MDA-MB-231**	0.07	1.3	>3.7
**HT-29**	89.3 ± 12.2	42.9 ± 16.3	83.3 ± 17.4				
**HeELA**	46.4 ± 4.3	86.8 ± 2.3	20.1 ± 4.2				
**5-FU**	**24 h**	**48 h**	**72 h**		**24 h**	**48 h**	**72 h**
**HEK-293**	44.6 ± 7.2	19.8 ± 2.3	14.5 ± 3.2	**HEK-293/HELA**	>1.1	0.6	0.9
**PNT1-A**	23.1 ± 4.55	21.2 ± 2.2	17.5 ± 7.6	**HEK-293/HT-29**	0.9	0.4	0.7
**MDA-MB-231**	42.6 ± 2.56	37.9 ± 12.3	21.0 ± 4.3	**HEK-293/MDA-MB-231**	1.0	0.5	0.6
**HT-29**	47.3 ± 1.2	23.4 ± 2.1	19.3 ± 1.2				
**HELA**	37.8 ± 4.3	31.2 ± 2.3	14.7 ± 0.6				

^a^ Means of three independent experiments.

**Table 2 ijms-22-06098-t002:** Primer sequences used for RT-PCR.

Primer	Forward (5′-3′)	Reverse (3′-5′)
**CASPASE 3**	GAGCACTGGAATGTCATCTCGCTCTG	TACAGGAAGTCAGCCTCCACCGGTATC
**CASPASE 8**	CATCCAGTCACTTTGCCAGA	GCATCTGTTTCCCCATGTTT
**CASPASE 9**	ATTCCTTTCCAGGCTCCATC	CACTCACCTTGTCCCTCCAG
**CASPASE 12**	GCCATGGCTGATGAGAAACCA	TCGCATCCCCAAAAGGTCAA
**CA9**	AGTCATTGGCGCTATGGAGG	TCTGAGCCTTCCTCAGCGAT
**NRF-2**	TTCGGCTACGTTTCAGTCAC	TCACTGTCAACTGGTTGGGG
**BAX**	TCCATTCAGGTTCTCTTGACC	GCCAAACATCCAAACACAGA
**BCL-2**	ATCGTCGCCTTCTTCGAGTT	ATCGTCGCCTTCTTCGAGTT
**LC3**	ATCATCGAGCGCTACAAGGG	AGAAGCCGAAGGTTTCCTGG
**BECLIN-1**	CGACTGGAGCAGGAAGAAG	TCTGAGCATAACGCATCTGG
**GAPDH**	GGAAGGACTCATGACCACAGT	GGATGATGTTCTGGAGAGCCC

**Table 3 ijms-22-06098-t003:** Primer monoclonal antibodies and seconder antibodies used for the Western Blot.

Antibody	Brand	Dilution Ratio	Rabbit/Mouse	Time
**Pro-Caspase 3**	ST John’s Laboratory	1/1000	Rabbit	Overnight +4 °C
**Cleaved Caspase 3**	ST John’s Laboratory	1/1000	Rabbit	Overnight +4 °C
**Cleaved Caspase 8**	ST John’s Laboratory	1/1000	Rabbit	Overnight +4 °C
**Cleaved Caspase 9**	ST John’s Laboratory	1/1000	Rabbit	Overnight +4 °C
**Cleaved PARP**	Milipore	1/1000	Rabbit	Overnight +4 °C
**CAIX**	Cell Signaling	1/1000	Rabbit	Overnight +4 °C
**B-actin**	Cell Signaling	1/50,000	Mouse	1–2 h +4 °C
**AntiRabbit IgG HRP**	Cell signaling	1/10,000	Rabbit	1–2 h +4 °C
**AntiMouse IgG HRP**	Santa cruz	1/10,000	Mouse	1–2 h +4 °C

## Data Availability

The data presented in this study are available on request from the corresponding author. The data are not publicly available due to privacy and/or ethical concerns.

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
