# Peer review of "Inhibition of Carbonic Anhydrase IX Promotes Apoptosis through Intracellular pH Level Alterations in Cervical Cancer Cells"

_ijms, 2021, doi:10.3390/ijms22116098_

Round 1

Reviewer 1 Report

This manuscript is a follow-up of the previous work of the authors, who synthesized a panel of sulfonamide derivatives as inhibitors of carbonic anhydrase, with potential anticancer effect. In this work, they have selected one of these compounds and characterized its effect thoroughly, using a wide range of methods.

Although the data presented by the authors are valuable and may enhance further research in the fields of carbonic anhydrases and anticancer drugs, the manuscript is not well written and presentation of the data is far from flawless. Before considering the manuscript for publication, following points should be addressed:

Line 47: “... investigated, how this compound triggers apoptosis and metastasis…” looks contradictory.

Lines 91-93: aerobic and anaerobic processes seem to be mixed up.

Line 114: what are “drugs targeting organic molecules”?

Line 139: the abbreviation of 5-fluorouracil should be explained.

  • 2.4.: it would be more convenient for the reader to present the data as a table or a graph.

Fig.4: it is uncelar, what the x axis represents. Also the text related to Fig.4 (lines 194-196) is difficult to understand.

  • 2.10: Gene Expression. The authors may consider making the title of this section more specific, e.g. expression of apoptosis-related genes. The same applies to the § 2.11. Protein Expression.

Line 431: the plural of the word “medium” is “media”.

Author contribution: does C.T. actually mean C.T.S.?

Acknowledgements: the authors incorrectly put abbreviations to this section. In fact, there are many unexplained abbreviations in the manuscript. They should be explained for the readers’ convenience. But not in the Acknowledgemets section.

In conclusion, while I am positive with respect for the results presented in the manuscript, I cannot recommend it for publication in the present form. The way the manuscript is written is rather sloppy. Besides the points mentioned above, the language of the manuscript has to be significantly improved.

Author Response

We have previously submitted our article entitled “Inhibition of Carbonic Anhydrase IX promotes apoptosis through intracellular pH level alterations in Cervical Cancer Cells” to be considered for publication in your journal International Journal of Molecular Sciences and received your letter including the comments of the reviewers. We want to express our deepest thanks to you and the reviewers as your comments helped us to improve the scientific quality of the manuscript. The article was revised and rewritten according to the comments of the reviewers. The sections of the manuscript were revised and modified. We look forward to receiving your answer and comments about the new version of our manuscript.

Sincerely yours,

Reviewer 2 Report

To summarize the manuscript entitled “Inhibition of Carbonic Anhydrase IX promotes apoptosis through intracellular pH level alterations in Cervical Cancer Cells” by Temiz et al., evaluating the effect of CAIX inhibitor (compound E) triggers apoptosis and metastasis in three cancer lines. They show that compound E exert the lowest IC50 and the highest selectivity index in CAIX-positive HELA cells. CAIX inhibition changes the morphology of HELA cells and increases the rate of apoptotic cells, and also this inhibition dramatically disturbed the homeostasis of intracellular pHi, MMP and ROS levels. There are some concerns that should be addressed by the authors before acceptance for publication.

Following are some concerns which need to be addressed by the authors:

  1. Please clarify the abstract. The objectives should be clearly addressed.
  2. Include a better rationalization for choice of compound E.
  3. The authors should add the potential role that CAIX may play in cervical cancer (PMID: 33803236; PMID: 26622782; PMID: 29725249; PMID: 21223596).
  4. In Figure 1, multiple groups are being compared please analyze the data using a mixed-ANOVA with a repeated measure post hoc test. Student’s t-test would be inappropriate.
  5. Figure 3 needs to be sharper.
  6. The limitation should been more fully developed in the discussion section.

Author Response

(The authors gave the same response as above.)

Reviewer 3 Report

The Authors present interesting findings in the area. However, a lot of errors occur throughout the text. The major concern is that some results and regulatory mechanisms might be more precisely and professionally described.

Most of the study was performed using only one cell line. I would recommend some further experiments to increase the significance of their findings or the Authors should explain why they used only one cell line

The manuscript needs extensive editing. Major editing of English language and style is highly required. The language should be corrected by a qualified reviewer of English.

Major revision is required, but in my opinion, the manuscript may be interesting for the Readers

The Authors should better address their findings. Authors present some interesting facts on proapoptotic action of the compound in cancer cells, but more extensive explanations for presented results should be given. The Materials and Methods and Results are poorly presented.

Kind regards

Some other points were listed below

Many English style and grammar mistakes occur in the MS, it should be improved, e.g.:

(153) is a staining type whose dual staining indicates the presence of an apoptotic cell.

(line 207 ) “The fact is that this increase occurred in proportion to the decrease in pHi prevented malign cell transformation (Figure 5).” – the meaning of the sentence is not clear

(321) “Treatment strategies that inhibit CAIX gene expression, characterized by intracellular pH change, have continued to be tested”. – the meaning of the sentence is not clear

(564)  4.15. Statistical Analysis “The distribution of data was controlled by … “(calculated using?)

Abstract:

line 46: “Here, we SYNTHESIZED a CAIX inhibitor (compound E) and investigated HOW this compound triggers apoptosis and metastasis in THREE cancer lines” – the Authors use the compound which has been synthesized and published before. Was it clear that compound E would elucidate apoptosis? Maybe it would be more proper: “and investigated whether this compound may trigger apoptosis and metastasis in cancer cells”. Besides: was apoptosis measured in three lines or just in HeLa cells? How did the Authors assess metastasis process? Maybe they should rather refer to cytotoxicity and ROS studies?

Introduction:

The Authors mention the role of HIF in cancer cells – it suggests that experiments will be performed in hypoxic conditions and the interaction HIF-CAIX will be studied here. Please, consider focusing rather on the meaning of CAIX in tumors and its role as a potential therapeutic target.

The statement (113) that CAIX …. causes metastasis is too far, even “in brief”

More info about compound C should be given  here, the Authors do not even refer in Intro to their former studies on its structure and properties – if the compound may inhibit CAIX in vitro – it should be given here as a justification to undertake the present study

Materials and Methods:

Please, define what “selectivity index” means

Please, check carefully and name properly company/country of equipment in this section (e.g. “fluorometry device (Spectramax, M5)”). The same for software (Li-Cor Fc).

ATCC designations for each cell line should be added

Please, justify why normal cells were not implemented for the study

All methods should be describe in similar way – here some are described more extensively and some were less detailed (4.2). The preparing of the cells for each experiment should be given (which line(s), number of the cells etc., for how long  the cells were incubated with the compounds, what was the control in this experiment, p value – for which groups)

The titles of Materials and Methods sections are confusing

e.g. 4.13. Gene Expression – do you mean “RNA extraction and real-time quantitative PCR”?, 4.14. Protein Expression – do you mean “Immunoblot analysis”?

The same in Results section:

2.3. Acridine Orange/Ethidium Bromide (AO/EB) Staining, 2.4. Annexin V-PI Staning, 2.6. Intracellular ROS Levels, 2.7. Mitochondrial Membrane Potential (MMP) (it would be Materials and Methods section title) Please, consider changing it for e.g. “compound E disrupts mitochondrial membrane potential in HeLa cells culture”

Results:

Please consider change of captions, e.g. “2.1. In Vitro Cell Viability Assay in Cells” -maybe for: “Compound C decreases viability of cancer cells?”

(126) „WST-1 method” – „WST-1 assay”?

Figures captions are too brief. The relevant information (e. g. name of the cell line) and the essential observations should be pointed out (e.g. Note that compound E decreased the expression of …..protein in….cells comparing to ….). It should be explained what “control” means

(211 ) Figure 5. Reactive oxygen species (ROS) levels detected by spectrofluorimetric (ASSAY?). The cells were stained with the ROS red dye and spectrofluorimetric (ASSAY?)was performed to determine the intracellular ROS. – the same info in the two sentences, no info about the experimental design, the effect of the compound

(238) Figure 7 – what “WT” means?

Please, add full names for acronyms, such as 5-FU, also in table/ figures captions

Lines 172-177 – please find a better way to show these results (you may show raw data in appendix) – are these numbers the same as results as in Fig 3? If yes, they should be presented only once, possibly with visible statistical analysis ()

(203) “The intracellular pHi and intracellular free radical (ROS) are MARKERS of CELL DIVISION  and MALIGNANT TRANSFORMATION” this is too much simplified, such statements must be changed throughout the test

(216, section 2.7.) please check whether MMP is a proper acronym for Mitochondrial Membrane Potential

Discussion

The Authors have performed several experiments which confirmed proapoptotic activity of compound E. However, the finding might be better discussed

The Authors should propose an explanation why compound E at 10uM was more proapoptotic than in higher concentrations

The Authors should not refer to p53 level/TP53 expression since it was not studied here

References:

The text need editing since another font type was used

Line 581 Acknowledgments (do you mean Abbreviations?): CAIX: Carbonic anhydrase IX; pHi: Intracellular pH; ROS: Reactive oxygen species; MMP: Mitochondrial membrane potential; AO/EB: Acridine orange-Ethidium 582 bromide; HIF-1: Hypoxia inducible factor-1.

Author Response

(The authors gave the same response as above.)

Round 2

Reviewer 2 Report

I think the authors respond the requests properly.

Author Response

We have previously submitted our article entitled “Inhibition of Carbonic Anhydrase IX promotes apoptosis through intracellular pH level alterations in Cervical Cancer Cells” to be considered for publication in your journal International Journal of Molecular Sciences and received your letter including the comments of the reviewers. We want to express our deepest thanks to you and the reviewers as your comments helped us to improve the scientific quality of the manuscript. The article was revised and rewritten according to the comments of the reviewers. The sections of the manuscript were revised and modified. We look forward to receiving your answer and comments about the new version of our manuscript.

Reviewer 3 Report

Some English language grammar mistakes still occur throughout the text (e.g line 43 “IN solid tumours”). However, the Authors put a lot of effort to improve the quality of the MS.

Comment 26: The different kind of font type was used in References section.

The manuscript is suitable for the publication, the listed points can be improved.

Kind regards,

Author Response

(The authors gave the same response as above.)
